# Syngas Production Improvement from CO2RR Using Cu-Sn Electrodeposited Catalysts

**DOI:** 10.3390/ma18010105

**Published:** 2024-12-30

**Authors:** Daniel Herranz, Santiago Bernedo Biriucov, Antonio Arranz, Juan Ramón Avilés Moreno, Pilar Ocón

**Affiliations:** 1Departamento de Química Física Aplicada, Universidad Autónoma de Madrid (UAM), C/Francisco Tomás y Valiente 7, 28049 Madrid, Spain; daniel.herranz@uam.es (D.H.); santiago.bernedo@estudiante.uam.es (S.B.B.); pilar.ocon@uam.es (P.O.); 2Departamento de Física Aplicada, Universidad Autónoma de Madrid (UAM), C/Francisco Tomás y Valiente 7, 28049 Madrid, Spain; antonio.arranz@uam.es

**Keywords:** CO_2_ reduction reaction, electrocatalysis, electrodeposition, Cu, Sn, bimetallic catalyst

## Abstract

Electrochemical reduction of CO_2_ is an efficient and novel strategy to reduce the amount of this greenhouse-effect pollutant gas in the atmosphere while synthesizing value-added products, all of it with an easy synergy with intermittent renewable energies. This study investigates the influence of different ways of combining electrodeposited Cu and Sn as metallic elements in the electrocatalyst. From there, the use of Sn alone or with a small amount of Cu beneath is investigated, and finally, the best catalyst obtained, which has Sn over a slight Cu layer, is evaluated in consecutive cycles to make an initial exploration of the catalyst durability. As a result of this work, after optimization of the Sn and Cu-based catalysts, it is possible to obtain more than 60% of the organic products of interest, predominantly CO, the main component of syngas. Finally, this great amount of CO is obtained under low cell potential (below 3 V), which is a remarkable result in terms of the cost of the process.

## 1. Introduction

The emission of large amounts of carbon dioxide (CO_2_) derived from human activities, such as fossil fuel combustion, industrial waste, or land use, is among the principal causes of global climate change [1]. CO_2_ is the most abundant greenhouse gas responsible for more than 50% of the global temperature increase. In order to achieve a sustainable society, great efforts are needed to approach a carbon-neutral energy cycle [2]. The main strategies comprise (i) reduction of CO_2_ emissions, (ii) carbon capture and storage (CCS), and (iii) carbon capture and utilization (CCU). The last approach focuses on utilizing CO_2_ itself as a carbon feedstock, thus transforming it into a source of high value-added products. From all the options explored in the literature to accomplish this goal, the electrochemical reduction reaction of CO_2_ (CO2RR) under mild conditions is one of the most attractive, driven simply by electricity that might be obtained from renewable sources [3]. Among the possible products of CO2RR, carbon monoxide (CO) is currently one of the most techno-economically viable [4] and is a relevant industrial feedstock that can be further processed and converted into high value-added compounds through well-established technologies like the Fischer–Tropsch (FT) process [5]. A minimum faradaic efficiency to CO of 33%, reaching a molar ratio CO:H_2_ of 1:2, is required for the syngas to be used in methanol synthesis or FT process [6]. Extended use of CO2RR to synthesize CO and other value-added products can potentially have an important impact on decreasing CO_2_ emissions.

There are three different possible CO2RR electrolyzer configurations [7], namely, H-cell, flow cell, and membrane electrode assembly cell (MEA-cell). The first two present a liquid electrolyte layer between the electrodes and the ion exchange membrane, allowing for the insertion of a reference electrode, but in general, they suffer from high ohmic drop and low energy efficiency. On the other hand, the MEA-cell forms a zero-gap structure with a thin film between the cathode and anode, significantly reducing ohmic resistance and improving energy efficiency [8]. Much of the research in the field has been conducted using a direct gaseous CO_2_ supply to the cathode interphase due to the high selectivity to CO achieved [9,10,11], but recently, the use of CO_2_ through solutions with carbonates and bicarbonates has attracted great interest. With these solutions, CO_2_ can reach the cathode surface mainly in two ways: first, by direct adsorption of dissolved aqueous CO_2_ on the cathode surface where the weakly alkaline KHCO_3_ inhibits the dissolution of CO_2_ into CO_3_^2−^ and maintains a relatively stable partial pressure of CO_2_ saturation, ensuring a relatively stable CO_2_ concentration [7]; and second, by direct CO_2_ generation at the membrane-cathode interface due to the reaction of bicarbonate ions and protons produced by the bipolar membrane [12,13], but usually, high overpotentials are required for the water splitting responsible for the proton presence [14,15]. The main advantages of using a CO_2_ feed within the liquid phase are that it significantly reduces the amount of CO_2_ present in the gaseous output mixture, thereby achieving higher CO_2_ utilization rates [16] and that it can be easily coupled to an alkaline-based carbon capture system by direct use of the CO_2_ capture solution [17].

The reactions involved in the CO2RR electrolyzer are the following:

Cathode:(1)$$CO2RR:\;\;\;{CO}_{2}+H_{2}O+2e^{-}\to CO+2OH^{-}$$
(2a)$$HER\;\left(acidic\right):\;\;\;2H^{+}+2e^{-}\to H_{2}$$
(2b)$$HER\;\left(alkaline\right):\;\;\;2H_{2}O+2e^{-}\to H_{2}+2OH^{-}$$

Anode:(3a)$$OER\;\left(acidic\right):\;\;\;2H_{2}O\to O_{2}+4H^{+}+4e^{-}$$
(3b)$$OER\;\left(alkaline\right):\;\;\;4OH^{-}\to O_{2}+{2H}_{2}O+4e^{-}$$

The electrocatalysts used for CO2RR can be prepared following several strategies, for instance, co-precipitation [18,19], chemical etching [20,21], hydrothermal synthesis [22,23], and electrodeposition [24,25]. Among them, electrodeposition offers the advantage of being an environmentally friendly technique, has high efficiency, and requires simple equipment. In addition, metals are introduced via salt precursors, which are abundant and suggest a low-cost approach in terms of obtention of the catalyst. On top of that, it allows for a wide use of 3D architecture substrates, and the addition of binders and conducting agents can be avoided. Moreover, the electro-synthesized catalysts can present several advantages for CO2RR: large electroactive surface area and fast charge/mass transfer rate due to the 3D architecture electrodes, precise structures and morphologies of the deposits due to the controlled growth process, and improved stability of the catalysts coming from the strong attachment to the substrate. Most of the electro-synthesized catalysts are either single metal catalysts or bimetallic catalysts. Examples of single metal catalysts are those synthesized with Pd, Sn, and Bi, whose main product is usually HCOOH [26,27,28], or others with Zn, Ag, and Au, whose main product is CO [29,30,31]. Surface-modified Cu electrodes have shown selectivity for HCOOH, CO, CH_4,_ and C_2+_ products like ethene, demonstrating a great diversity of options [32,33,34,35]. Bimetallic catalysts have attracted interest in recent years owing to their distinctive electronic and chemical properties compared to monometallic catalysts, which in many cases show higher efficiency due to the synergistic effect of the components. They have shown great potential for CO2RR, but their composition, structure, and understanding are still challenging. Among the metals used in bimetallic catalysts, Cu and Sn have attracted significant interest. Both are non-precious metals and, combined with other metals, have shown good CO2RR performance. For example, Wang et al. [36] prepared Cd-Cu catalysts by electrodeposition of Cd over Cu oxides at −10 mA·cm^−2^ for 1 h and used them at CO2RR with continuous CO_2_ gas supply at −8 mA·cm^−2^ and at −1.0 V versus RHE obtaining a high CO faradaic efficiency (FE) of 84%. Li et al. [37] synthesized a Cu-Sn bimetallic alloy that showed a high FE to HCOOH of 91.38% at −0.8 V vs. RHE and studied how the proportions of the metals affected the particle size and morphology of the catalysts. Due to the interest in this composition, other authors have prepared and tested Cu-Sn bimetallic catalysts for CO2RR [38,39,40,41], but most of them use an H-cell configuration with continuous gas feed to the cathode electrode.

If we focus on the reaction product of CO2RR to CO, electroreduction that occurs on the catalyst surface is a two-proton and two-electron transfer process. This reaction is, in fact, the simplest of all the CO2RR pathways. Based on the literature, the most selective catalysts for the evolution to CO are gold and silver [42,43], and for this reason, many studies in the literature focus on understanding the reaction mechanisms involved on the surface of these two metals.

Syngas production has multiple applications and arouses much interest in research studies. In particular, silver and gold are two metals known as excellent catalysts for obtaining high selectivity and efficiency toward CO [44,45,46,47,48]. Gold, more expensive than silver, shows greater selectivity and efficiency. Since gold is approximately 50–70 times more expensive than silver [49], there is much interest in developing silver-based catalysts for obtaining CO with high efficiencies. To produce organic compounds, copper is an interesting option to produce C_1_ and C_2_ organic compounds such as hydrocarbons and alcohols at significant current densities, as shown in the works of Hori et al. [42,43] and Azuma et al. [48], among others. Electrodeposited zinc on copper foil or single-atom catalysts based on Zn are interesting alternatives for the reduction of CO_2_ to CO or methane, respectively [50,51]. Moreover, Quin et al. have published very interesting results regarding work with Zn-based catalysts where they studied the influence of the ratio between the Zn (002) and Zn (101) facets, the latter being the one that prefers the CO2RR to CO, while the 002 facet favors the HER [52]. Finally, Stojkovik et al. [41] propose a very interesting approach in line with the circular economy development since the cathodic catalysts are based on the reuse of industrial waste Cu−Sn bronze, obtaining good CO efficiencies.

In this work, we propose different bimetallic catalysts with Cu-Sn compositions, synthesized by electrodeposition of the metals, varying their order and amount to assess the influence of these parameters. The catalysts were prepared by chronopotentiometry and then tested with the same operation mode for CO2RR in a zero-gap flow cell fed in the cathode side with a continuous flow of KHCO_3_-saturated CO_2_ solution, which is considered closer to a possible industrial application. The performance was evaluated at −25 to −200 mA·cm^−2^ current density range and 20 to 120 mL·min^−1^ electrolyte flow rates by quantifying the obtained products, the amount of CO_2_ in the outlet, and the average voltage measured during the reaction.

## 2. Materials and Methods

### 2.1. Reagents and Preparation Procedure of the Catalysts

The following reagents were used without further purification: KHCO_3_ (99%) and ethylenediaminetetraacetic acid (EDTA) were purchased from Scharlab (Barcelona, Spain), nickel foam (99.99%) from Nanografi Nano Technology (Ankara, Turkey), KOH (85%) from Labbox (Barcelona, Spain), Fumasep FAA-3-PE-30 membranes from FuMa-Tech (Bietigheim-Bissingen, Germany), and GDL carbon cloth with carbon MPL and treated with PTFE from Fuel Cell Store (Bryan, TX, USA), Gaseous compressed CO_2_ (99.7%) was purchased from AirLiquide (Madrid, Spain), CuCl_2_·2H_2_O (99.99%), and SnCl_2_·2H_2_O (99.99%) from Panreac (Barcelona, Spain) and Quality Chemicals (Esparreguera, Spain), respectively.

The electrolyte used for the electrodeposition process was a solution of 1 M KHCO_3_ (to increase conductivity), 0.01 M of EDTA, and 0.05 M CuCl_2_ as Cu precursor and/or 0.05 M SnCl_2_ as Sn precursor. The electrodeposition was performed in a three-electrode custom-made electrochemical cell where the carbon cloth GDL was pressed against a stainless-steel plate (used as current collector) at the bottom of the cell and in contact with the electrolyte. The carbon cloth acts as a working electrode, a graphite carbon rod as a counter electrode, and the electrode Ag/AgCl (KCl sat.) as a reference. The process was controlled by an Autolab PGSTAT302N potensiostat/galvanostat (Metrohm, Madrid, Spain), and the electrodeposition was performed by chronopotentiometry at −14 mA·cm^−2^ for different times depending on the catalyst (details are included in Table 1 and Appendix A). In Cu-Sn catalysts, Cu was electrodeposited over the carbon cloth, and afterward, the Sn was electrodeposited over the Cu layer, so Cu-Sn means “Sn over Cu”. The same applies to Sn-Cu, just changing the order of the metals. In the catalyst of Cu + Sn, both metals were electrodeposited at the same time, and in Cu*-Sn, a thin layer of Cu was first deposited for 2 min and then Sn was deposited on top. The electrodeposited catalysts were thoroughly rinsed with distilled water after the deposition to remove the electrolyte from the cloth and then dried in an oven at 60 °C overnight. The catalytic load was calculated using the mass difference of the empty and electrodeposited carbon cloth, ranging between 2 and 6 mg·cm^−2^ depending on the catalyst.

### 2.2. Structure Characterization and Morphology

Optical microscopy images were obtained using an Olympus BX41 from Evident (Barcelona, Spain) with 5×, 10×, 20× and 50× magnifications.

X-ray photoelectron spectroscopy (XPS) analysis was carried out in an ultra-high vacuum chamber at a base pressure lower than 1 × 10^−9^ mbar. A hemispherical analyzer (SPECS Phoibos 100MCD-5) from SPECS (Berlin, Germany) and Al Kα (1486.6 eV) radiation from a twin anode (Al–Mg) X-ray source operating at a constant power of 300 W were used. No charging effect was observed except for the “Cu + Sn mix” sample. For this sample, the binding energy was determined by referencing the C 1s peak of the C-C species at 285.0 eV to agree with the binding energies observed for the C-C species in the other samples. Before introducing the samples in the system, they were dried at 60 °C under a vacuum (10 mbar) for 3 h.

Raman spectra were recorded with a BWS415 i-Raman coupled to a BAC151B microscope both from BWTEK (Plainsboro, NJ, US) equipped with 20× and 50× objectives and a 532 nm laser as excitation light. All the spectra were recorded with the 50× objective, laser powers from 0.70 to 3.5 mW, and integration times between 5000 and 20,000 ms.

### 2.3. Flow Cell Setup, Electrochemical Measurements, and Product Analysis

The electrochemical flow cell used (ElectroChem Inc., Raynham, MA, USA) contained two graphite flow plates pressed together by two current collector plates with a gold coating. Silicon and PTFE gaskets were placed between the flow plates to avoid electrolyte and gas leakage and delimitated an active area of 2 × 2 cm. The electrolytes were driven by a dual-channel peristaltic pump D-25V from Dinko Instruments (Barcelona, Spain). Anolyte was an aqueous solution of 0.5 M KOH, and catholyte was composed of aqueous 0.5 M KHCO_3_ and 0.01 M EDTA, the latter being used to remove trace metal impurities [53]. A 3 × 3 cm Fumasep FAA-3-PE-30 membrane from FUMATECH BWT GmbH (Bietigheim-Bissingen, Germany) was used to separate anodic and cathodic compartments and Ni foam was chosen as anode catalyst, while the electrodeposited carbon cloth was the cathode catalyst. Gaseous CO_2_ was introduced in the catholyte tank by a syringe to saturate the solution with dissolved CO_2_. The complete scheme of the experimental setup is depicted in Appendix A.

The electrode double layer capacitance (C_DL_) was measured using cyclic voltammetry (CV) in a non-faradaic region of potentials at various scan rates 5–100 mV·s^−1^ (see Appendix A).

The electrochemical measurements were carried out using an Autolab PGSTAT302N potensiostat/galvanostat from Metrohm (Madrid, Spain). Currents between −25 and −200 mA·cm^−2^ and electrolytes flow rates between 20 and 120 mL·min^−1^ were carried out during CO2RR for at least 5 min each time to produce enough volume of gaseous products to extract them. The gas samples were extracted from the gas trap in the system using a 5 mL SGE gas-tight syringe from Fisher Scientific (Pittsburgh, PA, USA) and the products were analyzed using a GC (Varian 3900 with Carboxen-1006 PLOT Column) from Análisis Vínicos (Tomelloso, Spain) coupled to an MS (Pfeiffer Vacuum Hi-Cube) from (Tecnovac, Alcobendas (Madrid), Spain) with Argon as the carrier gas. A ramp from 35 °C to 245 °C with 30 °C·min^−1^ slope was employed in the GC for the adequate detection of the products. The obtained amounts of different compounds were recalculated and normalized to have the CO_2_ outlet as % of the total gas volume on one side and the CO2RR products on the other side. The reaction products of CO_2_, mainly H_2_ and CO, were renormalized to 100% as faradaic efficiencies. H_2_ quantification is not plotted with CO and CO_2_ in the figures for better clarity. In addition, the reaction products of CO2RR, H_2_, and CO were normalized to 100%. The sum of the % of CO, H_2_, and CO_2_ is greater than 100% since the first two correspond to FEs and the third to % of the compound in the outlet.

The faradaic efficiency was calculated as:(4)$${FE}_{i}=\frac{z_{i}\cdot F\cdot x_{i}\cdot n}{Q}$$
where $z_{i}$ and $x_{i}$ are the number of electrons transferred and the molar fraction, respectively, of the product i (z_i_ is 2 for CO, 2 for H_2_, 8 for CH_4,_ and 12 for C_2_H_4_); F is the Faraday constant (96,485.33 C·mol^−1^); n is the total number of mols of the products; and Q is the charge. The liquid phase in the cathode was monitored using the GC-FID/TCD technique without finding the presence of any organic compound from CO2RR. In addition, the anolyte solution was monitored without finding any signal associated with organic compounds. Since no products were detected in the liquid phase, all the reported products correspond to the gaseous phase.

## 3. Results and Discussion

The electrodeposition order of the selected metals in the catalyst has an obvious influence on the performance of CO2RR. To study different options, the catalysts were prepared with (a) first Cu and then Sn electrodeposition (Cu-Sn), (b) first Sn and then Cu electrodeposited on top (Sn-Cu), and (c) both metals electrodeposited at the same time from a solution containing both of them (Cu+Sn).

The synthesis of all catalysts was carried out by metal electrodeposition on a carbon cloth. The conditions and the CP curves can be seen in Table 1 and Appendix A, respectively. To check the adequate metal coverage of the surface, the electrodeposited carbon cloths were observed under an optical microscope. The images are presented in Figure 1 and show homogeneous coverage with some rifts due to the conductive carbon layer present in the cloth. Prior to the Cu-Sn catalyst design and optimization, we checked the performance of a catalyst based on Cu electrodeposition over the carbon cloth. The results showed a poor efficiency toward CO (<5%) and a reasonably good CO2RR efficiency toward methane (>20%) and ethene (~10%). Considering that our main goal is to obtain syngas, Cu-based catalysts were discarded in the context of this work. The thickness of the naked and overall carbon cloth + electrodeposited metals was measured with a digital caliper resulting in no detectable variation between naked and electrodeposited cloths, which is logical due to the relatively small amount of metals.

With the aim of testing the catalyst performance of CO2RR, first, a constant electrolyte flow rate of 80 mL·min^−1^ was maintained and different reduction currents were investigated. The CO2RR CPs conditions can be found in Appendix A and an example in Appendix A. The results are presented in Figure 2 and Appendix A.

Appendix A shows that the resulting reduction potentials are in the order of −2 V, which is a good value compared to the literature [54,55,56]. The values became more negative at higher reduction currents due to the faster reaction process demanded by the system. From the results included in Appendix A, the main reaction products in the gas phase are H_2_ and CO, with a general trend in the first two catalysts (Cu-Sn and Sn-Cu) of higher CO % at less negative reduction currents (Figure 2). These results show that for these catalysts, the lower reaction currents help promote the CO2RR vs. the HER, reaching high CO_2_ to CO conversion values of 34% or 31% with Cu-Sn and Sn-Cu, respectively. Additionally, it can be observed that the CO_2_ outlet values, which indicate the CO_2_ adsorbed in the surface but desorbed before reacting, show a general trend of decreasing % with more negative reduction currents. The highest values are obtained with the catalyst of Cu + Sn, while the lower CO_2_ outlet % values are obtained with the Cu-Sn catalyst. Based on the % of CO_2_ outlet, a low % may indicate a better performance of the catalyst (at a similar % of CO obtained), which in this case is achieved by the Cu-Sn catalyst. In general, % of CO_2_ outlet was lower than 50% for the four reduction currents tested in this work, reaching very low values (<40%) for the Cu-Sn catalyst. Based on the % conversion of CO_2_ to CO, the best catalyst was also Cu-Sn, obtaining 34% of CO at −25 mA·cm^−2^.

The electrolyte flow rate is another important parameter that has a high impact on the amount and proportion of obtained products [55]. Different electrolyte flows were tested with a constant current density of −25 mA·cm^−2^, from the optimum results explained above. The results are presented in Figure 3 and Appendix A.

The results show that the catalysts Sn-Cu and Cu + Sn obtain slightly lower CO% values at increasing electrolyte flows, whereas the Cu-Sn catalyst shows an important increase, reaching a high value of 46% of CO in the reaction products. The increase in CO_2_ outlet at higher flow values could be due to better adsorption of the molecule when the flow rate is higher (probably more turbulent), but only the Cu-Sn catalyst translates that to a better % of CO. Moreover, the increase in the CO_2_ outlet at higher flow rates is probably due to an easy desorption process.

Raman spectra of the surface of the catalysts were recorded before and after the use of CO2RR (Figure 4).

Cu appears as Cu^+^ and Cu^2+^, the first at 150 and 220 cm^−1^ and the second approximately at 300 cm^−1^. Sn can be found as Sn^2+^ at 112 and 210 cm^−1^ and as Sn^4+^ at 113, 200, 470, and 630 cm^−1^ [57,58]. Raman analysis confirms the presence of several Cu and Sn oxides in the bulk of the catalyst layers whose near-surface region is analyzed in detail by XPS.

The results of the superficial composition of the catalysts, obtained by XPS, are presented in Figure 5 and Appendix A and Table 2, comparing the composition of the samples “As prepared”, prior to their use in the flow cell for CO2RR, and “Used”, after the CO2RR process. The identification was performed according to the literature [59,60,61]. In Figure 5a, there is a clear trend of lower Sn % compared to Cu after the use of the catalysts; however, both Cu and Sn were partially removed from the carbon cloth surface, as could be visually observed, concluding that the Cu has a better adhesion to the microporous carbon surface than Sn in all the samples. This is a relevant result since it may be one of the major sources of catalyst degradation, and thus, for further improvement of these types of catalysts, the Sn adhesion and retention will be critical. The high initial percentage of superficial Sn in Cu + Sn comes from the preferential electrodeposition of Cu compared to Sn, which causes Cu to deposit first in the inner layer and Sn second and with relatively more abundance at an external layer [41]. From Figure 5b, a trend can be observed in all the samples to increase the content of lower oxidation state species of Cu (Cu^0^ and Cu^+^) after CO2RR compared to more oxidized Cu^2+^, which was expected since the processes taking place in the electrode are the reduction of CO_2_ and H_2_O, and some of the electrons will be derived to reduce Cu in the catalyst. On the other hand, in Figure 5c–e and Table 2, Sn shows a different behavior, only lowering the amount of the more oxidized species Sn^4+^ in the Cu-Sn catalyst. In Sn-Cu and Cu + Sn, the % of Sn^4+^ increases after the CO2RR process. The explanation might be related to the relative position of the metals in the catalyst. In Cu-Sn, Sn is the external and more exposed metal, and thus, the reduction process will occur mainly on its surface, while in the Sn-Cu one, Sn is not that much exposed and it may also transfer electrons to the superficial Cu, oxidizing itself in the process. The Cu + Sn catalyst shows behavior in between—it shows a higher percentage of Sn^4+^ after CO2RR like Sn-Cu but also an increase of Sn^2+^ like Cu-Sn. Since both metals are electrodeposited from the same solution, there was probably a higher interaction between them, allowing for more oxidation of Sn releasing electrons to Cu. This is coherent with the low initial percentage of Sn^0^ in the Sn-Cu catalyst where the underlying Sn would be mostly oxidized during the electrodeposition of Cu. Furthermore, it should be noted that the Cu + Sn catalyst obtained by electrodeposition of both metals at the same time did not present the formation of any alloy, at least on the surface. XPS analysis did not reveal the presence of Cu-Sn bond signals.

Based on the previous results, in order to discriminate the role of Cu under the Sn, Cu*-Sn and Sn catalysts were prepared. The Cu*-Sn catalyst was similar to the previous Cu-Sn but with a very small amount of Cu electrodeposited by a sort time of 2 min (vs. 20 min of the Cu-Sn catalyst), while the catalyst Sn only had this metal electrodeposited on the carbon cloth. The electrodeposition parameters and obtained voltages can be found in Table 1. The performance of the catalysts at CO2RR is presented in Figure 6 and Appendix A.

As observed in Figure 6a, an increase in the current density leads to a lower CO percentage in the products, like in the previous cases in this investigation. However, a higher maximum value is obtained, with 60% CO at −25 mA·cm^−2^ using Cu*-Sn, compared to the 34% CO of Cu-Sn catalyst in Figure 2. The Cu*-Sn catalyst also demonstrates improved performance compared to a pure Sn (100%) catalyst and the previous ones when varying the electrolytes flow at a constant low current density. It reaches values as high as 62% CO in the products at 120 mL·min^−1^. On the other hand, Sn shows a low maximum at 80 mL·min^−1^ and then goes down. The percentage of CO_2_ outlet presents a similar trend compared to the previous catalysts, decreasing with increasing current density (see Figure 6a) and increasing with the electrolyte flow (see Figure 6b).

To investigate a possible relation between the electrochemical active surface area of the electrodes and the performance, the double layer capacitance (C_DL_) was measured, (Appendix A). The results show a possible tendency of larger C_DL_ in the samples with higher Cu content among those composed of Cu and Sn. However, a direct relationship between C_DL_ and CO2RR performance could not be established, which is an indication of the higher relevance of the specific selectivity of the catalyst sites compared to the overall electrochemical surface area.

Cu*-Sn catalyst was studied after CO2RR by XPS, and the results are presented in Table 2 and Appendix A. A higher Cu content was observed in the Cu*-Sn catalyst than in Cu-Sn. This result was not expected since Cu*-Sn was supposed to have a small amount of Cu compared to Cu-Sn catalyst. After a detailed analysis of the results, we reached the following conclusion: the Cu*-Sn catalyst was used in four stability tests while the Cu-Sn catalyst was used only once. As we discussed, the catalysts suffer from an evident lack of adhesion of the metals to the carbon cloth and we observed a loss of material after its use in the CO2RR. It is clear that the Sn is lost in a higher proportion compared to Cu. For this reason, we believe that the Cu*-Sn catalyst lost more surface mass than the Cu-Sn catalyst, so it lost more Sn in comparison, hence it showed a higher proportion of Cu. The results also show an increased amount of Cu^0,+^ compared to Cu^2+^, which might be related to the initially smaller amount of Cu, and so a higher % of it was reduced over CO2RR. Those small differences could partially be the reason for the improved performance of Cu*-Sn, so further research gradually varying the amount of Cu will be needed in the future.

These results demonstrate the importance of the slight layer of electrodeposited Cu under the Sn to obtain high CO%. The good results obtained in this initial exploration of Cu*-Sn motivate the reuse of the synthesized catalyst to test stability. Some cycles of CO2RR were applied using the same catalyst, and the performance is presented in Figure 7 and Appendix A. Moreover, the synergistic effect by which Sn improves CO2RR to CO in the presence of a small amount of Cu is a plausible hypothesis and has been studied in other catalysts, for example, in the ORR [62].

In the previous results, a general trend of increase in CO_2_ outlet % could be detected with increasing electrolyte flow or decreasing current density. However, in this case, since these parameters are constant in each series, there is no common trend. Regarding the CO%, a clear decay is observed at −25 mA·cm^−2^ with the number of uses, reaching only 24% in the fourth use. At −50 mA·cm^−2^, the performance is not affected severely in the first three uses, even increasing slightly, and finally reaching 7% in the fourth use. This effect might be explained by a loss of active sites on the catalyst due to removal by electrolyte flow or manipulation (the cell was disassembled and assembled between uses). The loss of electrodeposited metal was visually observed, with a clear difference between the carbon cloth surface at the first and last use. The loss of metal in the surface would decrease the maximum reachable conversion from CO_2_ to CO (at −25 mA·cm^−2^) but might not be affected that much at higher current densities where most of the current was derived to HER.

Interestingly, the surface changes provoked by the degradation of the catalyst also led to an increase in the detected ethene, as can be observed in Appendix A.

Overall, when the results are compared to the latest literature, most of the studies work with H-cells where the CO_2_ is introduced directly into the solution in the cathode chamber with a high probability of reaching the cathode surface not only as dissolved CO_2_ but also directly as small gas bubbles, substantially improving the obtention of CO_2_-derived products and reducing the HER. FE to CO in those works is usually around 80–90%, but most of the time, the CO_2_ outlet is not informed. Due to the absence of direct CO_2_ gas bubbles reaching the cathode, this work is more comparable to those of carbonate/bicarbonate feed (see Table 3), for which Li and Shao have published a recent review [63]. They show that some catalysts are able to obtain high FE to CO as 82% and 60% in the studies of Lees et al. [54] and Zhang et al. [55], respectively, but the cell voltages needed are −3.4V and −3.7V in the mentioned studies. Most of the catalysts that obtain CO as the main product with concentrated bicarbonate solutions are based on Ag, while other studies have worked with catalysts based on Cu or Sn, but they have shown selectivity for other compounds [64,65], also working with high cell voltages. Similar voltage values have also been obtained in previous works by our group [56,66], which are produced by the use of bipolar membranes and their need for higher overvoltage for the water splitting, required to obtain protons next to the cathode surface to produce in situ CO_2_. The study that produces CO at a lower voltage (−2.2 V) [67] works with a cation exchange membrane and obtains 15% FE_CO_. Compared to those results, our work reaches a high FE_CO_ of 62% with a small cell voltage of −2.0 V and an acceptable 49% CO_2_ outlet, demonstrating very promising results.

## 4. Conclusions

This study investigates bimetallic catalysts for CO2RR synthesized by different electrodeposition strategies of Cu and Sn. First, the combination of Sn over Cu shows a better performance of CO production, reaching 46% with 44% CO_2_ outlet, compared to Cu over Sn or a mixed electrodeposition. The XPS analysis of the catalysts before and after use revealed that Cu has better adhesion than Sn to the carbon cloth surface and a clear difference in the oxidation states before and after the CO2RR process. It is evident that the catalyst degradation issue must be addressed by, for example, improving the electrodeposition conditions, such as using lower voltages or currents and longer times. The influence of reducing current density and electrolyte flow rate was analyzed, showing a clear preference for smaller values of the first and a varying behavior of the second. Furthermore, Sn and Sn over a slight amount of Cu catalysts were synthesized and tested, demonstrating that it is not only Sn but its interaction with an adequate amount of Cu that gives an improved FE to CO, reaching 62% with 49% CO_2_ outlet at −25 mA·cm^−2^, 120 mL·min^−1^ electrolyte flow, and −2.0 V cell voltage, a very promising result due to high values at relatively low voltage in a MEA-flow cell fed only with dissolved CO_2_ aqueous bicarbonate solution. Moreover, an in-depth theoretical study of the CO2RR pathways to CO with different catalysts would be desirable in order to better understand the synergistic effect between Sn and Cu. In addition, the optimal amount of the first Cu layer electrodeposited seems to be a critical parameter and this issue needs to be investigated in more detail to get the best performance of the Cu-Sn catalyst toward CO.

## Figures and Tables

**Figure 1 materials-18-00105-f001:**
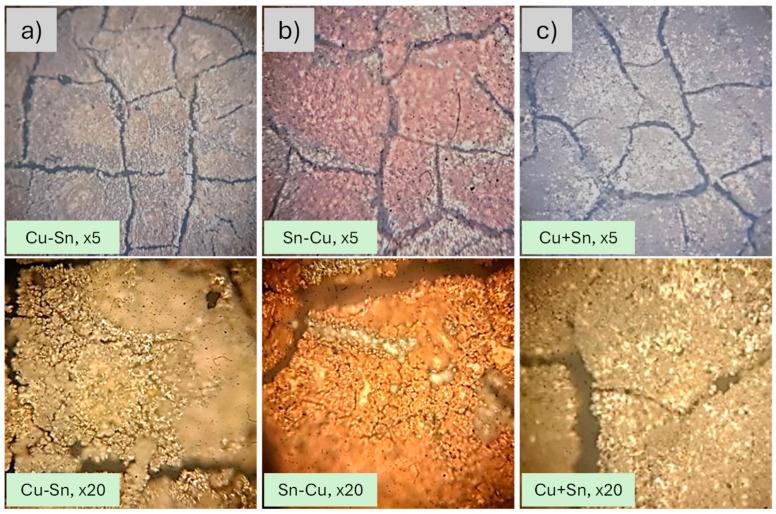
Microscopy pictures of the catalysts after electrodeposition for the Cu-Sn (**a**), Sn-Cu (**b**) and Cu+Sn (**c**) materials. The differences in intensity of colors are due to ambient light.

**Figure 2 materials-18-00105-f002:**
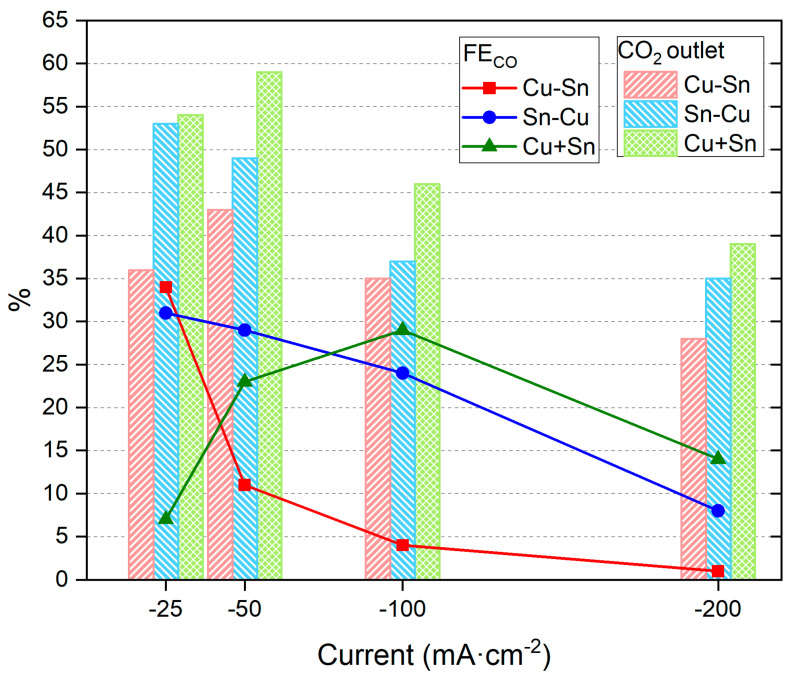
CO_2_ outlet % and % FE_CO_ of catalysts Cu-Sn, Sn-Cu, and Cu + Sn at a constant flow rate of 80 mL·min^−1^ and different reduction currents. Red, blue, and green symbols indicate FE to CO for the three tested catalysts. Red, blue, and green bars indicate % of CO_2_ outlet for the three tested catalysts.

**Figure 3 materials-18-00105-f003:**
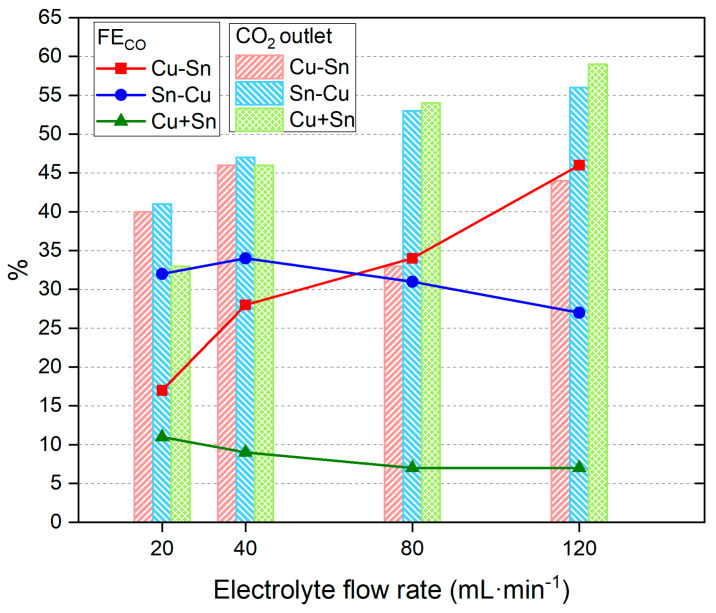
CO_2_ outlet % and % FE_CO_ of catalysts Cu-Sn, Sn-Cu, and Cu + Sn at a constant current density of −25 mA·cm^−2^ and different electrolyte flow rates. Red, blue, and green symbols indicate FE to CO for the three tested catalysts. Red, blue, and green bars indicate % of CO_2_ outlet for the three tested catalysts.

**Figure 4 materials-18-00105-f004:**
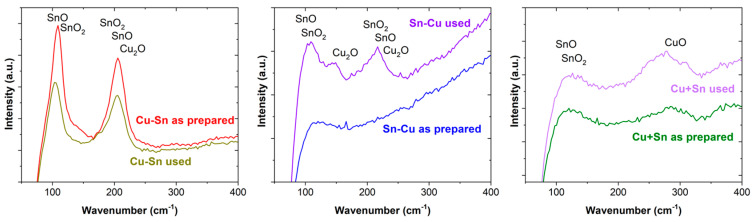
Raman spectra before and after the use of the catalysts for CO2RR.

**Figure 5 materials-18-00105-f005:**
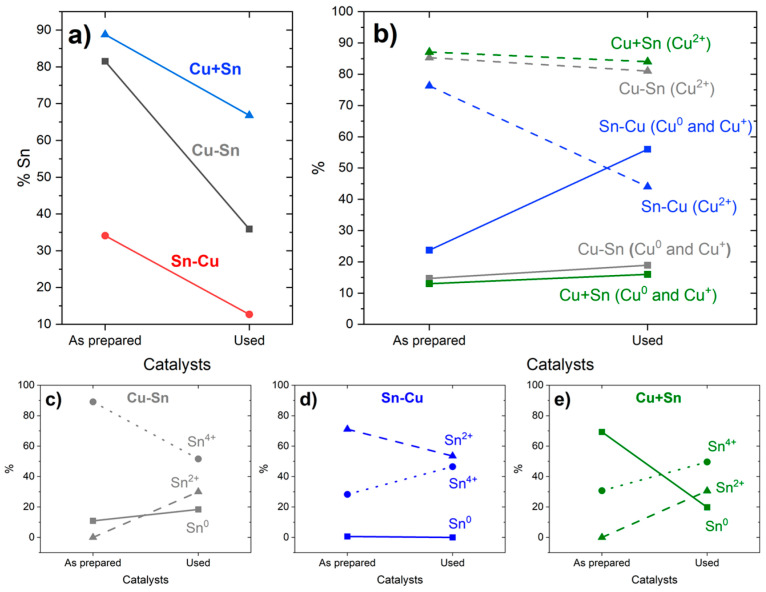
A variation of metallic percentage of catalysts “As prepared” and “Used” (after CO2RR) considering (**a**) Sn in total Sn and Cu, (**b**) Cu species in total Cu, and (**c**–**e**) Sn species in total Sn.

**Figure 6 materials-18-00105-f006:**
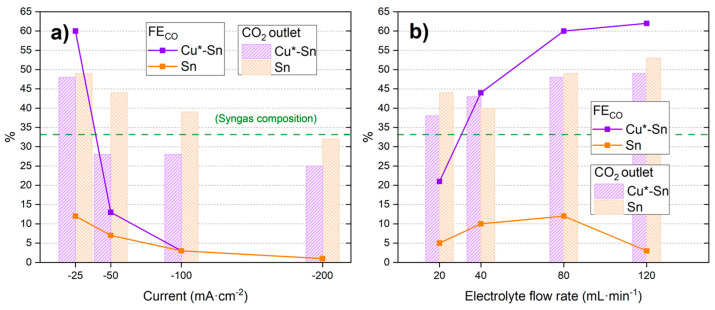
CO_2_ outlet % and % FE_CO_ of catalysts Cu*-Sn and Sn at (**a**) constant electrolyte flow rate of 80 mL·min^−1^ and (**b**) constant current density of −25 mA·cm^−2^.

**Figure 7 materials-18-00105-f007:**
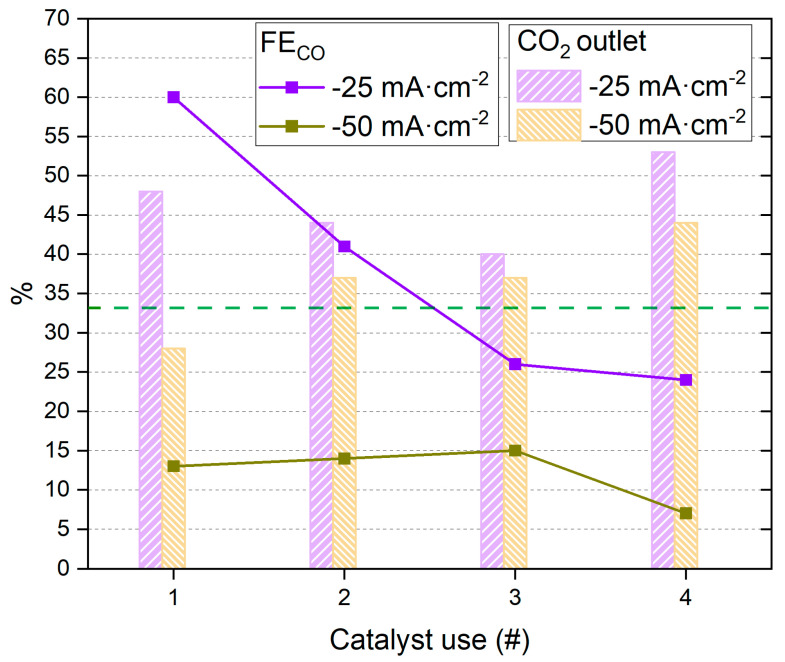
CO_2_ outlet % and % FE_CO_ of catalysts Cu*-Sn at 80 mL·min^−1^ after increasing the number of catalyst uses. For reader’s clarity, syngas composition is shown (green dotted line).

**Table 1 materials-18-00105-t001:** Electrodeposition conditions of the metals to prepare the catalysts. Potential values are vs. Ag/AgCl (KCl sat.) the reference electrode.

Catalysts	Metal Layers	Current (mA·cm^−2^)	Time (min)	Potential (V)
Cu-Sn	1st, inner layer: Cu	−14	20	−1.3
	2nd, outer layer: Sn	−14	20	−1.5
Sn-Cu	1st, inner layer: Sn	−14	20	−1.9
	2nd, outer layer: Cu	−14	20	−1.0
Cu+Sn	Cu+Sn	−14	20 (×2)	−1.8
Cu*-Sn	1st, inner layer: Cu	−14	2	−1.3
	2nd, outer layer: Sn	−14	20	−1.5
Sn	Sn	−14	20 (×2)	−1.5

**Table 2 materials-18-00105-t002:** Composition of metallic percentage in catalysts “As prepared” and “Used” (after CO2RR) determined by XPS considering total Cu and Sn (“Cu and Sn” columns), Cu at various oxidation states (“Cu” columns) and Sn at various oxidation states (“Sn” columns).

Catalyst	Cu and Sn		Cu		Sn		
	% Cu	% Sn	% Cu^0^ and Cu^+^	% Cu^2+^	% Sn^0^	% Sn^2+^	% Sn^4+^
Cu-Sn as prep.	19	81	15	85	11	0	89
Cu-Sn used	64	36	19	81	18	30	52
Sn-Cu as prep.	66	34	24	76	1	71	28
Sn-Cu used	87	13	56	44	0	54	47
Cu+Sn as prep.	11	89	13	87	69	0	31
Cu+Sn used	33	67	16	84	20	31	50
Cu*-Sn used	74	26	25	75	16	31	53

**Table 3 materials-18-00105-t003:** A summary of the latest studies using carbonate/bicarbonate feed.

Cathode Catalyst	Catholyte	FE (%)	CO_2_ Outlet (%)	Current (mA·cm^−2^)	Cell Voltage (V)	Reference
Cu*-Sn electrodeposited	CO_2_ (g) sat. in 0.5 M KHCO_3_ and 0.01 M EDTA	62 (CO)	49	−25	−2.0	This work
Ag composite	3 M KHCO_3_	82 (CO)	not reported	−100	−3.4	[54]
Porous Ag	3 M KHCO_3_	60 (CO)	not reported	−100	−3.7	[55]
Ag nanoparticles	2 M KHCO_3_	46 (CO)	41	−200	−3.8	[66]
Electrodeposited Ag	2 M KHCO_3_ with 0.02 M DTAB	85 (CO)	50	−100	−3.5	[56]
Ag foam	3 M KHCO_3_	15 (CO)	~45	−500	−2.2	[67]
SnO_2_ nanoparticles	3 M KHCO_3_	58 (HCOO^−^)	not reported	−100	−4.1	[64]
Cu foam	3 M KHCO_3_ with 3 mM CTAB	27 (CH_4_)	not reported	−400	−7.2	[65]
Cu-Sn bronze	CO_2_ (g) with 0.1 M KHCO_3_	85 (CO)	not reported	−6	−0.8 V vs. RHE	[41]

## Data Availability

The original contributions presented in this study are included in the article/Appendix A. Further inquiries can be directed to the corresponding author.

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
