# Peer review of "Syngas Production Improvement from CO2RR Using Cu-Sn Electrodeposited Catalysts"

_materials, 2024, doi:10.3390/ma18010105_

Round 1
Reviewer 1 Report
Comments and Suggestions for Authors
Please find additional comments in the attached document.

The English could be improved to more clearly express the research.
Reviewer 2 Report
Comments and Suggestions for Authors
Please see the report

Reviewer 3 Report
Comments and Suggestions for Authors
In their study, the authors have investigated the use of electrodeposited bimetallic catalysts with Cu-Sn compositions, synthesized by electrochemical deposition. The catalysts were tested for CO2RR in a zero-gap flow electrolyzer.
The discussions were conducted appropriately, and the data are suitable for publication after addressing the following issues:
- Figure 2, and 3 Y axis is not clear for the reader!
- When The electrodeposition happends in two-step, perhaps the second metal to be deposited in the responsible of the catalytic acticity? has the authors tried to test the Cu only?
- EDX would enrich the discussion especially to determine the distribution of the elements on the surface when doing two-step electrodeposition? It is hard to know if the first element deposited is exposed to CO2 !
- What is the thickness of the films in different conditions?
- When electrodepositing Sn and Cu at the same time (one step), what is the composite/phase formed?
- Any liquid products have been detected?
- The FEs are the exact values or they are normalised to attain 100%?
Round 2
Reviewer 1 Report
Comments and Suggestions for Authors
The authors have addressed my questions. No additional comment will be provided.
Reviewer 3 Report
Comments and Suggestions for Authors
The revised manuscript can be accepted for publication!